# Methodological aspects of economic evaluations conducted in the palliative or end of life care settings: a systematic review protocol

Claudia Fischer ,[1] Eva Chwala,[2] Judit Simon[1]

[1]Health Economics, Medical University of Vienna, Center for Public Health, Vienna, Austria
[2]University Library, Medical University of Vienna, Vienna, Austria

**Correspondence to**
Dr Claudia Fischer;
claudia.fischer@meduniwien.ac.at

## ABSTRACT

**Introduction** In light of this growing palliative care and end of life care patient population, as well as new (expensive) drugs and treatments, quality research providing evidence for decision-making is required. However, common research guidance is lacking in this field, especially in respect to the methods applied in economic evaluations. Therefore, the aim of the planned systematic review is to identify and summarise relevant information on methodological challenges, potential solutions and recommendations for conducting economic evaluations of interventions in adult patients, irrespective of their underlying disease and gender in the palliative or end of life care settings, with no restrictions in regards to countries/geographical regions. The results of this systematic review may help to clarify the current methodological questions and form the basis of new, setting specific methods guidelines and support ongoing applied economic evaluations in the field.

**Methods and analysis** A systematic review will be conducted using Medline, Embase, Health Technology Assessment Database and NHS Economic Evaluation Database to identify the studies published from 1999 onwards with relevant information on methodological challenges, potential solutions and recommendations for conducting economic evaluations in the palliative or end of life care settings. Articles in English, German, Spanish, French or Dutch language will be considered. Two independent reviewers will conduct the screening of articles; any discrepancies will be resolved by discussion and involvement of a third reviewer. Predesigned data extraction forms will be applied, consequently narratively synthesised and categorised. Studies' methodological quality will be critically appraised. Besides existing economic guidelines and checklists for specific information on the palliative and end of life care sector will be searched.

**Ethics and dissemination** Ethical approval is not required, as this is a planned systematic review of published literature. An article will be disseminated in a related peer-reviewed journal, as well as presented at leading palliative care and health economic conferences.

**PROSPERO registration number** CRD42020148160.

## Strengths and limitations of this study

► To the best of our knowledge, this systematic review, which will be conducted and reported according to the current highest methodological standard, is the first one to identify methodological aspects and challenges to consider when conducting an economic evaluation in the palliative or end of life care settings.

► A robust methodology including specific search strategy, tailored and pilot tested search strings developed in cooperation with an information specialist, has been applied to develop a targeted search for our broad research question.

► While our language restriction in the screening process might present a potential limitation, this shall be only marginal in the high-income setting considering also the good coverage of countries with a strong track record in economic evaluations and palliative/end of life care aspects.

► Although formal transferability assessment will be carried out as far as possible, a separate review tailored to low and middle-income settings may become necessary if the current review identifies relevant cultural/spiritual/preference aspects important also for economic evaluations.

► This systematic review will be carried out alongside a large EU project (the iLIVE project), which allows direct testing of our findings, develop practice-applied recommendations and external review of our findings by the iLIVE stakeholders.

## INTRODUCTION

Over the past century, patterns of mortality in the Western world have changed.[1] While sudden death, mainly due to infectious disease, used to be the major cause of death, nowadays there is an increase of the ageing population living with advanced stages of incurable chronic conditions such as cancer, chronic heart failure or chronic obstructive pulmonary disease near the end of life.[2 3]

It is crucial to enable this growing patient group to live well at the end of their lives as well as to die well.[4 5] This requires quality

research providing evidence for decisions on clinical guidelines, treatments and services.[6] In light of the new (expensive) drugs and interventions, an effective and efficient resource allocation is of great importance in the palliative and end of life care setting.[7] Studies have estimated the cost of care in the last year of life at 25%–30% of all medical expenditures during a lifetime.[8 9] This number is thought to even further increase in light of the ageing population and the prospect of the potentially reduced number of informal carers in the future.[10]

Economic evaluations are the comparative analysis of alternative courses of action in terms of both, their costs and consequences,[11] and have been widely applied in other fields to answer questions in regards to the cost-effectiveness of healthcare interventions and inform decision-makers and commissioners.[12] Nevertheless, the economic component has often been overlooked in palliative care or end of life care studies, leaving several unanswered questions for the quality assurance, monitoring, funding and evaluation of palliative care or end of life care.[13 14] Palliative and end of life care research lack common research guidance,[15] which would be essential as these settings differ from other healthcare fields in several aspects, which is why existing methods of economic evaluations may not be suitable for them. For example, compared with interventions in other fields, the ultimate focus of interventions in the palliative or end of life care settings is not on the extension of life but rather on the quality of dying, patient's dignity or strengthening relationships with loved ones.[16 17] Therefore, the appropriateness of generic quality of life (quality adjusted life years (QALY)) measures for economic evaluations in the palliative or end of life care settings may be questioned, as their measured quality dimensions may not reflect those dimensions relevant for palliative or end of life care patients.[18]

Further, it is well known that informal carers play an important role in the care of patients in many disease fields, such as chronic conditions.[19] The contribution of informal carers in the care of patients in the palliative or end of life care fields is thought to be substantially high.[20 21] Although the general acknowledgement of their importance, often their contributions, both in respect to costs and outcomes, are not included in economic evaluations.[22 23] This is due to diverse reasons, for instance, (1) payers want a perspective applied that reflects costs that emerge for them,[24] (2) there is no gold standard how to value (ie, attribute a monetary value to) indirect costs of informal care[25–27] and (3) ethical concerns about involving participants in data assessment and research at a suitable time, especially in regards to finances, which may add to the already enormous emotional and physical burden they face.[28] However, leaving these costs and outcomes unconsidered, may result in a biased reflection of the true costs and benefits of interventions in the palliative or end of life care settings.[18]

To generate valid and useful evidence and decision base, the research methods of economic evaluations

applied in the palliative care or end of life care fields should consider these specific conditions in their methodology. Therefore, this systematic review aims to summarise the methodological aspects and challenges (eg, valuation of informal care, the usability of QALYs or other outcome types in this field) and recommendations as well as potential solutions to overcome these problems and consequently provide methodological guidance when conducting economic evaluations in the palliative or end of life care settings, irrespective of the underlying disease, gender or geographical region.

## OBJECTIVES
The specific objectives of this study are as follows:
1. To identify methodological aspects and challenges of conducting economic evaluations in palliative or end of life care patients irrespective of their underlying disease, gender or geographical region.
2. To synthesise recommendations and potential solutions to overcome these identified challenges.
3. To sum-up methodological challenges, which have remained unresolved so far.
4. If feasible, to develop a methodological framework guideline for economic evaluations in the palliative and end of life care settings.

## METHODS AND ANALYSIS
### Protocol
This systematic review protocol has been developed according to the recommendations from the Preferred Reporting Items for Systematic Review and Meta-Analysis Protocols (PRISMA-P) 2015 statement[29] and PRISMA for abstracts checklist.[30] A PRISMA-P file and PRISMA for abstracts checklist is provided in online supplementary appendix 1. Besides, the review will follow the five-step approach on the state-of-the-art methodology for conducting systematic reviews of economic evidence by Van Mastrigt *et al*.[31]

The work on the described systematic review has been started in June 2019 (conceptualisation of the idea of the systematic review) and will take approximately until June 2021 (finalisation of the data synthesis and write up of results).

### Eligibility criteria
#### Population
Adult (a person older than 19 years of age) palliative or end of life patients, irrespective of the underlying disease, gender.

#### Comparator(s)
Not specified.

#### Context
Palliative and end of life care in different care settings such as hospitals, hospices, nursing homes or home of the patient. No explicit restrictions in regards to countries/

geographical regions are made. However, due to some necessary language restrictions, it is likely that the generalisability of the overall findings will be limited to Europe/North America/South America and Australia. Context specificity and transferability will be further assessed based on the findings.

## Main outcome(s)

The outcome of interest are methodological aspects and challenges (eg, valuation of informal care, the usability of QALYs or other outcome types in this field) for conducting economic evaluations in the palliative or end of life care settings.

## Study design

To serve the broad nature of our research question, we have decided a priori to develop a protocol that includes different study types published in peer-reviewed journals in this systematic review. Articles reporting on methodological aspects and challenges of conducting economic evaluations in the palliative or end of life care setting are suspected being:

► Systematic reviews (and meta-analysis): a structured search has been applied and transparent and explicit methodological criteria are used to select the included studies.
► Narrative reviews: including a qualitative summary and discussion of pivotal studies known to the subject experts.
► Observational (eg, case report or cohort), interventional studies.
► (Full) economic evaluations.
► Economic guidelines and checklists.
► (Editorial) discussions of the literature and commentaries: including contributions to theory building or critique, arguing the case for a field of research or a course of action or summaries of literature.
► Qualitative studies.

## Report characteristics

In this systematic review, no limit in regards to the setting will be applied, including clinical settings (eg, hospital and nursing homes), as well as community settings or the patient's home. Depending on whether we will be able to identify sufficient information per individual setting in which care may be provided, we will present the identified information clustered per individual care setting, otherwise, no distinction per setting will be undertaken. In the field of health economics, major methodological advancements have taken place in the past 10 years, which outdate methods that were commonly applied earlier. Moreover, palliative care practice itself has undergone significant changes since 2000.[32] For sensitivity reasons, we extended the time limit for publication 1 year before this date. All articles published after 1 January 1999 will be considered for inclusion. Articles written in English, German, French, Spanish or Dutch will be considered for inclusion.

The authors can read and analyse articles in English, German and Dutch. Further, the iLIVE project consortium includes native speakers who will provide help with articles of interest in Spanish and French.

## Information sources and search strategy

In collaboration with an information specialist, a systematic search in the electronic databases Medline, Embase, NHS Economic Evaluation Database and the Health Technology Assessment Database produced by the Centre for Reviews and Dissemination will be systematically conducted. Appropriate Medical Subject Headings terms will be selected and used and combined with free text words to develop a search strategy, which will then be translated to the different databases. The developed search strategy will be based on a review of published search filters for palliative care and health economics (Canadian Agency for Drugs and Technology in Health (CADTH) economic search filter https://www.cadth.ca/resources/finding-evidence/strings-attached-cadths-database-search-filters#eco, OVID Expert Searches Health Science http://resourcecenter.ovid.com/site/resources/expert_search/healthexp.html, CareSearch Palliative Care search filter https://www.caresearch.com.au/caresearch/tabid/377/Default.aspx, PubMed Health Sciences Research Queries https://www.nlm.nih.gov/nichsr/hedges/HSR_queries_table.html, Scottish Intercollegiate Guidelines Network (SIGN) Search filter https://www.sign.ac.uk/search-filters.html, McMaster Search filter https://hiru.mcmaster.ca/hiru/HIRU_Hedges_MEDLINE_Strategies.aspx).[33–37] Different combinations of search terms and specifications have been tested in regard to their sensitivity for our research question. For each applied modification, the first 100 articles have been evaluated to decide whether the modification is applied or not.

Detailed inclusion and exclusion criteria are presented in table 1.

The final search syntax for Medline (OVID) is a modification of the reviewed search strings of Gomes et al[37] and Rietjens et al[33]:

1. Palliative Care/
2. exp Terminal Care/
3. Terminally Ill/
4. palliat*.mp.
5. (terminal* adj6 (care or caring or ill or illness*)).ti,ab,ot,kf.
6. (end of life or last year of life or lyol or life's end).ti,ab,ot,kf.
7. advanced cancer.ti,ab,ot,kf.
8. Hospices/
9. hospice*.ti,ab,ot,kf.
10. bereave*.ti,ab,ot,kf,hw.
11. 1 or 2 or 3 or 4 or 5 or 6 or 7 or 8 or 9 or 10
12. exp *Health Care Costs/
13. ((health care or healthcare) adj3 cost*).ti,ot,kf,kw.
14. *'Costs and Cost Analysis'/
15. Cost-Benefit Analysis/mt [Methods]
16. exp models, economic/

**Table 1** Inclusion and exclusion criteria

| | Inclusion criteria | Exclusion criteria |
|---|---|---|
| Population | Adult* palliative or end of life patients, irrespective of the underlying disease, gender. | |
| Study design | Systematic reviews (and meta-analysis), narrative reviews, observational or interventional studies, (editorial) discussions and commentaries, (full) economic evaluations, economic guidelines and checklists, qualitative studies. | |
| Outcome | Methodological aspects and recommendations for conducting economic evaluations in the palliative and end of life care settings. | |
| Type of publication | Articles with available full text in English, German, Spanish, French or Dutch language. | (Conference) abstracts |

*Following the WHO definition, an adult is a person older than 19 years of age.[44]

17. (economic* adj3 (evaluat* or aspect* or health or analy* or model* or framework* or frame work* or method*)).ti,ab,ot,kf,hw.
18. economics.ti,ot,kf.
19. Palliative Care/ec
20. exp Terminal Care/ec
21. Hospices/ec
22. 12 or 13 or 14 or 15 or 16 or 17 or 18 or 19 or 20 or 21
23. 11 and 22
24. recycl*.ti,hw,kf,jw.
25. (waste or life cycle assessment).jw.
26. 24 or 25
27. 23 not 26
28. limit 27 to (dutch or english or german or french or spanish)
29. limit 28 to 'all child (0 to 18 years)'
30. limit 29 to 'all adult (19 plus years)'
31. 29 not 30
32. 28 not 31
33. limit 32 to yr='1999–2019'

The search syntaxes for other databases are presented in online supplementary appendix 2.

Besides, we will screen relevant websites (standard health economic evaluation associations (eg, International Society for Pharmacoeconomics and Outcomes Research (ISPOR), international Health Economics Association (iHEA)), the resources of the members area of the International Collaborative for Best Care for the Dying Person, as well as reports of health technology assessment (HTA) bodies (eg, National Institute for Health and Care Excellence (NICE)) to identify potential relevant guidelines to conduct economic evaluations in the palliative or end of life care settings, as well as a specific checklist for quality appraisal and value frameworks.

Furthermore, we will screen the articles for any potentially relevant article they refer to in the manuscript text.

## Study records
### Data management
For transparency and reproducibility a clear documentation of all searches, the electronic database searches, the references searched and the hand search gray literature, will be kept. The search syntaxes translated for all searched databases will be made available in the appendix of the systematic review, a PRISMA flow chart will be presented, illustrating the numbers of records retrieved and selection flow through the screening rounds. All identified references will be imported and combined in a single EndNote library. Duplicate records of the same reports will be removed.

### Selection process
Two screening rounds will be conducted. In the first screening round, the title and abstract of the articles will be screened. Two reviewers will independently screen all articles, the reviewers' independent decisions on inclusion and exclusion of articles will be compared and kappa statistics calculated. In case of the absence of an article's title/abstract, the full text of the article will be directly screened for potential inclusion. Those articles which are selected for further inclusion based on the title and abstract screening will undergo the second screening. In the second screening round, full texts will be screened and assessed for eligibility against the prespecified inclusion and exclusion criteria. Any disagreements will be discussed among the two, if the disagreement cannot be solved a third reviewer will be approached to reach a final decision. A PRISMA flow diagram[29] of the study selection process will be prepared to present an overview of the data collection process and the decisions that have been made in the course of it.

### Data extraction
For the data extraction from papers standardised data extraction forms will be created in Excel (see online supplementary appendix file 3 for illustration of categories). One researcher will extract the data with the second researcher independently checking the accuracy of the extracted information. Relevant missing information or unclear information will be dealt with by contacting original authors of included studies. To ensure that the data extraction form is properly applicable, it will undergo piloting, testing it for its user-friendliness and completeness on a subsample of the total number of retrieved included studies. If questions arise, these will be discussed and if necessary, the data extraction form will be revised accordingly.

### Data items
From each included article describing methodological aspects of economic evaluations in the palliative and end

of life care settings standard bibliographical information (ie, author, journal, publication year and country) and study design will be reported. Further, information on the aspect of economic evaluation described, challenge reported, the potential impact of the problem described, potential solution or recommendation described, will be abstracted. We will also collect information (eg, study characteristics and patient characteristics) from the identified applied economic evaluation. The assessed outcomes and cost categories considered in the economic evaluation will be abstracted and summarised.

## Data synthesis

A narrative synthesis of the included studies will be conducted following the Cochrane Collaboration guidelines[38] for which two data extraction sheets in Excel will be prepared, one for studies on methodological studies and one for applied economic evaluations. For the data extraction sheet for methodological studies, we a priori defined major economic evaluation characteristics by using the Consolidated Health Economic Evaluation Reporting Standards (CHEERS) checklist as a base.[39] For methodological/qualitative papers, such lists do not exist. Instead, we predefined a set of attributes in our data extraction sheet that we will collect as standard. Of course, these categories will remain open for modification and extension, depending on which information we will identify in the data extraction and analysis phase. If required, the categories will be refined or extended depending on the identified level of information and characteristic. The level of evidence for the described challenge, potential impact, concerns and solutions will be discussed per identified subtheme. With the help of the second data extraction sheet, which has been developed specifically for applied economic evaluations, the characteristics of the identified economic evaluations will be summarised as well as overviews of the considered outcomes and cost categories created.

For included articles, in which the specific country and hence the specific healthcare system is of relevance, the transferability of study findings will be assessed using transferability checklists.[40]

In case we can find sufficient information in regards to recommendations and potential solutions in respect to the challenges identified, we will develop a methodological framework guideline for economic evaluations in the palliative or end of life care setting in form of a critical appraisal checklist of good practice, aiming to improve the design, the collection of data (ie, outcomes and costs) as well as analysis of data in economic evaluations in this field. The methodological framework will emphasise on a set of recommendations, which differ from those when conducting economic evaluation in settings outside the palliative or end of life care settings.

The manuscript will be prepared following the PRISMA guidelines.

## Assessment of methodological quality and risk of bias

To be able to appraise the overall quality of evidence, and the strength of a conclusion drawn from it studies methodological quality will be critically appraised. Formal quality checks will be carried out where relevant lists are readily available (eg, NICE quality appraisal checklists,[41] CHEERS checklist[39] or PRISMA checklist[42]) and will record the quality/risk of bias information in our data extraction sheet. Two reviewers will be involved in the risk of bias assessment, one will fill in the quality appraisals, the second one will conduct quality checks, discrepancies will be solved by discussions and if necessary, consultation of a third reviewer. Once the extraction is finalised, we will decide on how this information can formally be incorporated into the synthesis of the evidence once data extraction is finalised and we have a clearer overview of the feasibility.

## ETHICS AND DISSEMINATION

The systematic review will synthesise information described in the published literature. Ethical approval is not required, as the proposed systematic review will not use primary data. This review will provide an overview of methodological aspects to consider when conducting an economic evaluation in the palliative and end of life care settings. Findings will be widely disseminated through peer-reviewed publication, conference presentation(s) and via other planned dissemination activities in the course of the EU project 'the iLIVE project—live well, die well, a research programme to support living until the end'.[43]

## PATIENT AND PUBLIC INVOLVEMENT

Our results and the developed framework will be undergoing a review process, that is, discussed and potentially refined with the experts within the iLIVE consortium and subsequently also with the palliative and end of life care community at international scientific conference discussions. Specifically, we will ensure to involve the iLIVE PP advisory board representatives, which is currently set-up and will consist of patients and public advisors, in the interpretation of the review findings. As recommended by Van Mastrigt *et al* these reviewers will not have been involved in the development of the framework and will include experts in the clinical area, methodological or HTA expert, or the targeted patient population.[31]

## DISCUSSION

To our knowledge, this will be the first systematic review, which will be conducted and reported according to the current highest methodological standard, to identify methodological aspects and challenges to consider when conducting an economic evaluation in the palliative or end of life care settings.

The systematic overview of the identified methodological aspects and challenges of conducting economic evaluations in the palliative and end of life care settings will help to deepen understanding of the challenge to conduct economic evaluations in these settings. The aim is to provide a systematic and comprehensive overview of the currently existing evidence base. For instance, provide an overview of the currently used outcome measures used in economic evaluations in the palliative and end of life care settings, provide a summary of their advantages and disadvantages, and the discussion whether or not generic measures may be used in these settings and provide an overview of available evaluation studies. The overview should help authors of future economic evaluations to recognise potential challenges and drawbacks in their study design. Moreover, it can play an important role in assessing the quality of evidence from previously conducted economic evaluations focusing on palliative and end of life care patients.

The second aim of this systematic review is to synthesise recommendations and potential solutions to overcome the identified challenges when conducting economic evaluations in the palliative or end of life care settings. These findings can help to clarify current methodological questions and present guidance for authors when setting up the methodology for future economic evaluations.

The summary of methodological challenges, which have remained unresolved so far is another aim of this systematic review and should guide researchers when planning and prioritising evaluation studies in regards to methodological questions of economic evaluations in the palliative and end of life care studies.

Further, recognising the absence of a comprehensive framework addressing the specific challenges of designing and executing economic evaluations in the palliative or end of life care settings, the results of this systematic review may form the basis of new, setting specific methods guidelines. Given we identify sufficient information in regards to recommendations and potential solutions in respect to the challenges identified, as well as guidelines and evaluation studies in this field, we will develop a methodological framework guideline for economic evaluations in the palliative or end of life care settings in form of a critical appraisal checklist of good practice. This will add to available general reporting guidelines, such as CHEERS.[39] In the other case, we will highlight the areas, which will need further attention and outline a research agenda, in order to be able to set up such a framework. With the proposed framework, it is aimed to improve and standardise the methodology and execution of future economic evaluations conducted in the palliative or end of life care settings, which should increase their comparability and overall transparency.

Our research question is quite broad with numerous challenges in the development of a targeted search strategy without this being too narrow and missing relevant articles neither being too broad resulting in an unmanageable amount of hits. Therefore, a robust methodology including a specific search strategy for multiple electronic databases of peer-reviewed literature, and tailored search strings carefully refined for every database searched and developed in cooperation with an information specialist have been developed and piloted.

Further, the screening process will be double screened by two individual researchers, considering articles published in English, German, Dutch, Spanish and French as eligible for inclusion. While this language restriction might present a potential limitation, this shall be only marginal in the high-income setting considering also the good coverage of countries with a strong track record in economic evaluations and palliative/end of life care aspects. Consequently, it is likely that the generalisability of the overall findings will be limited to Europe/ North America/South America and Australia. A separate review tailored to low-income and middle-income settings may become necessary if the current review identifies relevant cultural/spiritual/preference aspects important also for economic evaluations. Formal transferability assessment will be carried out as far as possible. The quality of a systematic review is highly dependent on its transparency of the methodology and reporting of results. To assure full transparency the systematic review protocol is reported according to the recommendations of the PRISMA-P statement. Further, the systematic review is registered with the International Prospective Register of Systematic Reviews. Also the systematic review itself will follow the recommendations of the PRISMA statement.

A final strength of this systematic review is that it will be carried out alongside a large EU project (the iLIVE project), which allows direct testing of our findings and develop practice-tested recommendations for this area. Further, the iLIVE project includes different relevant stakeholders, which enables a comprehensive external review of our findings. Specifically, we will ensure to involve the iLIVE PP advisory board representatives in the interpretation of the review findings.

**Acknowledgements** The iLIVE project – live well, die well, a research programme to support living until the end has received funding from the European Union's Horizon 2020 research and innovation programme under grant agreement No 825731.

**Contributors** JS and CF conceived the idea, developed the research question, study methods and data extraction form. EC and CF designed and developed the search strategy and strings. CF drafted the manuscript with contribution from JS. All authors approved the final manuscript.

**Funding** This systematic review is part of the iLIVE project – live well, die well, a research programmeto support living until the end which has received funding from the European Union's Horizon 2020 research and innovation programme under grant agreement No 825 731. The funders did not play a role in decisions in the development or publishing of the protocol.

**Competing interests** None declared.

**Patient and public involvement** Patients and/or the public were not involved in the design, or conduct, or reporting or dissemination plans of this research.

**Patient consent for publication** Not required.

**Provenance and peer review** Not commissioned; externally peer reviewed.

**ORCID iD**
Claudia Fischer http://orcid.org/0000-0001-7574-8097

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
