## [Reviewer comments · BMJ Open]

ARTICLE DETAILS

TITLE (PROVISIONAL)	Methodologic aspects of economic evaluations conducted in the palliative or end of life care settings: a systematic review protocol
AUTHORS	Fischer, Claudia; Chwala, Eva; Simon, Judit

VERSION 1 – REVIEW

REVIEWER	Clare Gardiner University of Sheffield, UK
REVIEW RETURNED	27-Nov-2019

GENERAL COMMENTS	Thank you for the opportunity to review this paper, the review protocol addresses an important and neglected topic and I'm glad to see attention being paid to the challenges of economic evaluation in palliative care. Unfortunately there are a number of serious issues with the manuscript, such that I don't think it is currently ready for publication. As it stands the review methods are not robust enough to warrant publication, and the rationale and impact of the work are not clearly defined. I've provided some more in depth comments below, I'm sure that with some greater consideration this protocol will be publishable and will contribute to the evidence base but in its current form I'm afraid I have to recommend it is rejected. The abstract needs to include the aim or objectives, and also a brief description of what the review hopes to achieve, or what outcomes you hope will result from this review. I don't think it's necessary to include ethics or dissemination in the abstract. The introduction gives a good overview of the topic but it's a little brief and the rationale for conducting this review could be made much stronger. Can you discuss in a little more depth why existing methods of economic evaluation are not suitable for palliative care? E.g. problems with the use of QALY's in palliative care, issues with valuing and costing informal carer contributions etc. Page 5 – under population you don't include primary research studies which is a little confusing as it seems from looking at your inclusion criteria that they are to be included. Page 6 - Under study designs you don't include qualitative studies. I'm not sure if this is an accidental omission (in which case please include) or whether this is intentional (in which case please provide a robust justification as I struggle to see why they should be excluded).
---

	Page 6, line 40 – The sentence beginning “In case enough information can be collected...” is a little unclear, and I’m not sure what is meant by individual setting in this context, can you clarify? One of the objectives is “If feasible, to develop a methodological framework guideline for economic evaluations in the palliative and end-of-life-care setting”. Please can you include details of how you will go about developing this framework somewhere in the methods? Please describe how data published in different languages will be handled, will you be translating? Page 9 under data synthesis - can you provide some more detail on this process, particularly how you will identify the economic evaluation characteristics, will these be decided a priori (in which case how) or will they be derived during the analysis process? Page 9 –quality assessment is not only for quantitative systematic reviews, there are a wide range of quality assessment tools for different study designs which will help appraise the overall quality of an evidence base, and the strength of conclusions drawn from it. I strongly suggest you re-consider and include some quality appraisal. Page 10 under risk of bias – This should include methods for assessing risk of bias of individual studies within the review. Page 10 patient and public involvement – it’s a shame you weren’t able to include patients/public in the design of this protocol as this would add much more weight to its value. I appreciate it is not always possible to include patients/public but if this is something you are able to do it would add value. The discussion is very brief and doesn’t provide much information on how the results of the review will be used. It would be really useful to engage with some of the wider literature on economic methods/pall care and discuss how your review will contribute to this evidence base. The discussion could be tied in much more directly to the research objectives, highlighting how these objectives can be met through your review and what the impact of the work will be, who will benefit, and what the consequences of the review will be. Minor issues: Table 1 – please define age range for adult There seems to be repetition of the PRISMA -P table which is included twice? Typo line 6, page 17
--	--

REVIEWER	Ghislaine van Mastrigt Maastricht University The Netherlands
REVIEW RETURNED	07-Dec-2019

GENERAL COMMENTS	General comments to authors Dear authors, I reviewed your study protocol entitled ‘Methodologic aspects of economic evaluations conducted in the palliative or end-of-life-care
--

	settings: a systematic review protocol'. This manuscript is clearly written and summaries a relevant topic. I have enjoyed reading it and have listed the comments and the suggestions for improvement for you. For details, see below. Comments for editor and authors Specific comments and recommendations for revision Abstract (page 2) Relevant information on the methods of the study are missing in the abstract, please add more details, for this use the PRISMA abstract guideline (http://prisma-statement.org/Extensions/Abstracts). Risk of bias assessment for this review is highly recommended; see also comments in section (Assessment of methodological quality). Strengths and limitations (page 3) No limitations are discussed only strengths. Introduction (page 4) The research question is quite broad; please revise using the suggested PICO (see below). Methods and Analysis (page 5) The eligibility criteria stated on page 5 could be improved as follows.  1. The population for this study is: "adult palliative and end-of-life care patients irrespective of underlying disease or gender" and not also the stated study designs 2. Comparators: does not need to be specified, as this is not relevant for this review. it is unclear what is meant by: "outside the palliative care" 3. Context: please specify which countries/geographic regions will be included in the review. 4. Main outcomes; please specify examples of methodological aspects and challenges. Add also other outcomes or synonyms like; recommendations, impact of problems, suggested solutions, concern and recommendations. In addition, add also "economic evaluations" as they are also important outcomes for this review. For details on methods of systematic reviews of economical topics, see three papers added to reference list of this review report. Study design (page 6): Guidelines and checklists are also relevant to add to the list. (Full) economic evaluations are will provide information that is more relevant and are therefore important studies to consider for this review. Observational and interventional studies are less relevant to include. Information sources and search strategy (pages 6-7) Besides searching for relevant studies in major databases, reports of HTA bodies are also be a relevant sources for finding information related to this topic (like NICE, http://guidance.nice.org.uk/ and Canadian Agency for Drugs and Technologies in Health (CADTH), https://www.cadth.ca). Searching for relevant citations (studies) can also be done by checking the references in known publications or by searching for
--	--

	additional studies that are cited in articles known to be relevant (such as Web of Science), this is also an essential step to find all relevant evidence. Search syntax (page 7 and appendixes) Please specify if peer reviewed, search strings were used. They to use them as much as possible; see https://sites.google.com/a/york.ac.uk/issg-search-filters-resource/). Adapt search strings using the suggestions for PICO and eligibility criteria. Where the search strategies already pilot tested? Please specify how details of searches will be recorded. Data management (page 8) Describe how data extraction sheets will be handled and which programs will be used for it (eg word, excel). Selection process (page 8) How will selection of title and abstract will be done when no abstract is available? For instance for editorial, comments, or guidelines. Please specify. Data items and appendix 3 General aspects The data extraction sheet (appendix 3) is very brief and contains no explanation/description of the items. This sheet could be improved by pilot testing it for a few studies. A picklist displaying varies answers in excel could decrease workload and could also improve consistency of answers. Specific aspects  • Not all studies will be published in a "journal" please add more answer options. • Please adapt and add more details, in item "study design" (e.g. economic evaluations (EEs), guideline and checklist). Aspects of EE described are only applicable for EEs. • Add open text space for all EE items. Information on: full or partial EE, Incremental cost-effectiveness analyses (ICERs) and analyses of uncertainty are also relevant to record. • What are relevant items for guidelines, editorials, reviews and all other type of publications selected? • Add more details/synonyms and examples for methodological problems/challenges, problems, concerns, recommendation and solutions. • For the most items, it would be also relevant to add a space in the sheet, for recording the specific page or section where the information was found by the reviewer in the document. • Finally, it is unclear why "Type of EE", "resource use" and "ethical implications" are bold. Assessment of methodological quality (page 9-10) Quality assessment of EEs or guidelines is very relevant for this review as it gives an indication of the methodological quality of the studies discussing the methodological challenges, which are the topic of interest for this review. Low quality studies/guidelines could have other challenges as good quality ones. On page 11, row 19 it is mentioned that the quality of evidence will be assessed, please explain how this will be done if you do not check the quality of the selected studies/guidelines (all types). Funding statement (page 13)
--	--

	Please state more clearly the name of the project. Reference list For improvement of this protocol, I would recommend to use the following paper series.  [ ] https://doi.org/10.1080/14737167.2016.1246960 [ ] https://doi.org/10.1080/14737167.2016.1246962 [ ] https://doi.org/10.1080/14737167.2016.1246961
--	---

REVIEWER	Dr Nikki McCaffrey Deakin Health Economics, Deakin University, Victoria, Australia
REVIEW RETURNED	15-Dec-2019

GENERAL COMMENTS	Please note, the views expressed below belong solely to this reviewer. The comments are intended to be constructive and the feedback aims to improve the quality of the manuscript. Generally, this is a well written and well-structured manuscript describing a systematic review protocol. The title adequately describes the study and the abstract is a fair summary of the study. The introduction clearly states the objectives of the study. Overall, the methods chosen appear relatively robust and appropriate for the stated objectives based on the information provided. However, given the review’s aims, I think “economic evaluations testing the efficiency of an intervention or economic evaluations in the palliative care settings” should be included in the review, as the discussion sections of such articles will provide useful information on methodological aspects and challenges of conducting economic evaluations in the palliative and end-of-life care setting. Omitting such studies would be a major limitation of the review. On a more minor note, there are a number of published and publicly available search filters for identifying palliative care research (see CareSearch as example https://www.caresearch.com.au/caresearch/tabid/377/Default.aspx). Why not use one of these validated search filters? Please clarify whether the two reviewers are independently screening the titles and abstracts (first screening round). How will the kappa statistic inform the process, e.g. what if the kappa value is low? Whilst the results of this systematic review should provide interesting findings, important information on methodological aspects and challenges of conducting economic evaluations in the palliative and end-of-life care setting may be missed due to the exclusion criteria.
---

VERSION 1 – AUTHOR RESPONSE

Reviewer #1 -----

R1.1. *The abstract needs to include the aim or objectives, and also a brief description of what the review hopes to achieve, or what outcomes you hope will result from this review. I don't think it's necessary to include ethics or dissemination in the abstract.*

Authors' Response	We agree with the reviewer's suggestion on the missing information in our abstract. However, the "ethics and dissemination" section is a required item of the abstract following the BMJ Open guideline.
Actions	Information on the systematic review's aim and the anticipated added value of it have been added to the abstract.

R1.2. *The introduction gives a good overview of the topic but it's a little brief and the rationale for conducting this review could be made much stronger. Can you discuss in a little more depth why existing methods of economic evaluation are not suitable for palliative care? E.g. problems with the use of QALY's in palliative care, issues with valuing and costing informal carer contributions etc.*

Authors' Response	We agree with the reviewer that the need for this systematic review can be emphasized in our "introduction" section.
Actions	The rationale and the need for conducting this planned systematic review has been extended in the "introduction" section, especially focusing on the difficulty to apply existing methods of economic evaluations in the palliative or end of life care settings.

R1.3. *Page 6 – under population you don't include primary research studies which is a little confusing as it seems from looking at your inclusion criteria that they are to be included.*

Authors' Response	We agree with the reviewer that primary research studies should also be stated in the examples listed.
Actions	Empirical studies (observational or interventional studies) have been added to the examples of listed studies relevant for inclusion in the "methods" section.

R1.4. *Page 6 - Under study designs you don't include qualitative studies. I'm not sure if this is an accidental omission (in which case please include) or whether this is intentional (in which case please provide a robust justification as I struggle to see why they should be excluded).*

Authors' Response	Thank you for noticing this omission from the list.
Actions	We have added "qualitative studies" to the study designs considered for inclusion.

R1.5. *Page 6, line 40 – The sentence beginning "In case enough information can be collected..." is a little unclear, and I'm not sure what is meant by individual setting in this context, can you clarify?*

Authors' Response	We were referring to the previous sentence, in which the different settings, in which palliative and end of life care can be provided, are listed, e.g. hospital, nursing homes, and patients home.
Response	We thought it would be of interest to separate the identified information for these settings. However, this will only be possible if we will be able to identify enough information per setting.
Actions	We clarified in the unclear sentence on the settings in the manuscript.

R1.6. *One of the objectives is "If feasible, to develop a methodological framework guideline for economic evaluations in the palliative and end-of-life-care setting". Please can you include details of how you will go about developing this framework somewhere in the methods?*

Authors' Response	We agree that more information on the potential development of the framework should be provided to the reader. We added more details, we will focus on potential differences in recommendations we have identified from economic evaluations conducting in settings outside the palliative or end of life care settings. The identified information will be synthesized and recommendations and a critical appraisal checklist, formulated, subsequently these will be reviewed by the iLIVE consortium members and the palliative and end of life care community at international scientific conference discussions.
Actions	More information on the potential development of the framework has been provided in the "methods" section. Also in the "discussion" section we added information on the framework, its potential impact and use.

R1.7. *Please describe how data published in different languages will be handled, will you be translating?*

Authors' Response	The authors are able to read and analyze articles in English, German, and Dutch. Further, the iLIVE project consortium includes native speakers who will provide help with articles of interest in Spanish and French.
Actions	This missing detail about the handling the articles in different languages have been added to the manuscript.

R1.8. *Page 9 under data synthesis - can you provide some more detail on this process, particularly how you will identify the economic evaluation characteristics, will these be decided a priori (in which case how) or will they be derived during the analysis process?*

Authors' Response	Using the Cheers (Consolidated Health Economic Evaluation Reporting Standards) checklist as a base, we defined major economic evaluation characteristics a priori, which are listed in the data extraction sheet and which will be used to cluster the identified information in the manuscript. For methodological/qualitative papers, such lists do not exist. Instead, we pre-defined a set of attributes in our data extraction sheet that we will collect as standard. Of course, these categories will remain open for modification and extension, depending on which information we will identify in the initial data extraction phase.
Actions	We added more details on how we identified the information in the "data synthesis" subsection of the manuscript.

R1.9. *Page 9 –quality assessment is not only for quantitative systematic reviews, there are a wide range of quality assessment tools for different study designs which will help appraise the overall quality of an evidence base, and the strength of conclusions drawn from it. I strongly suggest you reconsider and include some quality appraisal.*

Authors' Response	Our initial approach to quality appraisal was driven by the facts that due to the many different types of studies we are planning to review, no single standard quality checklist can be applied. Furthermore, it is currently not straightforward how the application of several different quality checklists can be compared and synthesized in one review. Nevertheless, we have reconsidered this approach based on the recommendation of both reviewers. We will indeed carry out formal quality checks where relevant lists are readily available (e.g. NICE Quality appraisal checklists, CHEERS checklist or PRISMA checklist) and will record the quality/risk of bias information. We have now added a quality assessment criteria to our data extraction sheet as well. We will then decide on
---

	how this information can formally be incorporated into the synthesis of the evidence once data extraction is finalized and we have a clearer overview of the feasibility.
Actions	The text in the manuscript has accordingly been modified. A section has been added to the data extraction sheet allowing to add the rating of the studies quality.

R1.10. Page 10 under risk of bias – This should include methods for assessing risk of bias of individual studies within the review.

Authors' Response	We now included the information on which checklists we will use for assessing the risk of bias of the different studies, which will be included in our systematic review. See also our response to R1.9.
Actions	The text in the section “assessment of methodological quality and risk of bias” has accordingly been modified in the manuscript. The quality appraisal checklists, which will be used, have been listed and cited.

R1.11. Page 10 patient and public involvement – it's a shame you weren't able to include patients/public in the design of this protocol as this would add much more weight to its value. I appreciate it is not always possible to include patients/public but if this is something you are able to do it would add value.

Authors' Response	Thank you, this is a very important point! We agree that the involvement of patients and the public is crucial for the weight of this reviews value. The draft protocol has been critically reviewed by the iLIVE consortium, on basis of which, several modifications have been executed. Further, the potentially developed framework will undergo an external review process, i.e. discussed and potentially refined with the experts within the iLIVE consortium and subsequently also with the palliative and end of life care community at international scientific conference discussions. We will also make sure to specifically involve the iLIVE PP advisory board representatives in the interpretation of the review findings.
Actions	We added information on the external review in the “patient and public involvement” section of our manuscript.

R1.12. The discussion is very brief and doesn't provide much information on how the results of the review will be used. It would be really useful to engage with some of the wider literature on economic methods/pall care and discuss how your review will contribute to this evidence base. The discussion could be tied in much more directly to the research objectives, highlighting how these objectives can be met through your review and what the impact of the work will be, who will benefit, and what the consequences of the review will be.

Authors' Response	Thank you for this comment. We have now added information per research aim in regards to the expected impact, potential users and applications, and final outputs beyond scientific publications to reflect better the importance of this review.
Actions	The discussion section has been revised and expanded according to the reviewer's suggestions.

R1.13. Minor issues:

Table 1 – please define age range for adult There seems to be repetition of the PRISMA -P table which is included twice?

Typo line 6, page 17

Authors' Response	We are afraid that we were not able to identify the error the reviewer refers in his comment. Nevertheless, we have now amended the relevant description of age in the manuscript and removed one table. We hope that these changes address the above comment as well.
Actions	The WHO definition for an adult, which we considered, has been added to the manuscript. One PRISMA-P table has been removed from the uploaded files.

Reviewer #2 -----

Dear authors,

I reviewed your study protocol entitled 'Methodologic aspects of economic evaluations conducted in the palliative or end-of-life-care settings: a systematic review protocol'. This manuscript is clearly written and summaries a relevant topic. I have enjoyed reading it and have listed the comments and the suggestions for improvement for you. For details, see below.

Authors' Response	We thank for the reviewer for the careful revision of our protocol and the very constructive and helpful remarks.
---

R2.1. Abstract (page 2)

Relevant information on the methods of the study are missing in the abstract, please add more details, for this use the PRISMA abstract guideline (<http://prisma-statement.org/Extensions/Abstracts>).

Risk of bias assessment for this review is highly recommended; see also comments in section (Assessment of methodological quality).

Authors' Response	We thank for the reviewer for this suggestion and applied the PRISMA abstract checklist to check whether all recommended items, which are applicable to a systematic review protocol, have been mentioned. Further, we agree with the reviewer on the importance of the quality assessment and will take quality appraisal into consideration when analyzing the included articles. See also our responses to R1.9 and R1.10.
Actions	The abstract has been modified according to the applicable PRISMA abstract checklist items. The PRISMA abstract checklist has been cited in the "methods" section of the manuscript. The "assessment of methodological quality and risk of bias" section has been modified, the quality appraisal, which will be applied described and tools listed.

R2.2. Strengths and limitations (page 3): No limitations are discussed only strengths.

Authors' Response	We agree with the reviewer and have elaborated on both the limitations and strengths of our planned study.
Actions	We modified the according text in the discussion section. Also, the strengths and limitations bullet points in the beginning of the manuscripts have been modified accordingly.

R2.3. Introduction (page 4)

The research question is quite broad; please revise using the suggested PICO (see below).

Methods and Analysis (page 5)

The eligibility criteria stated on page 5 could be improved as follows.

1. The population for this study is: "adult palliative and end-of-life care patients irrespective of underlying disease or gender" and not also the stated study designs
2. Comparators: does not need to be specified, as this is not relevant for this review. it is unclear what is meant by: "outside the palliative care"
3. Context: please specify which countries/geographic regions will be included in the review.
4. Main outcomes; please specify examples of methodological aspects and challenges. Add also other outcomes or synonyms like; recommendations, impact of problems, suggested solutions, concern and recommendations. In addition, add also "economic evaluations" as they are also important outcomes for this review.

For details on methods of systematic reviews of economical topics, see three papers added to reference list of this review report.

Authors' Response	We thank the reviewer for the suggestion to revise our PICO and accordingly our research question. Since we have now explicitly added applied economic evaluations to our review, we agree with the suggested revisions and have amended the text according. Regarding the country/geographic regions context, we will not limit our search, as we will not only screen for economic evaluations but different types of methodological studies, for which the country where it has been conducted should not be an exclusion criteria. However, due to some necessary language restrictions, it is likely that generalizability of the overall findings will be limited to Europe/North America/South America and Australia. We plan to assess context specificity and transferability based on the findings and will implement formal methods for references suggested in R2.12 where relevant. We would like to thank the reviewer for the suggested papers.
Actions	The PICO, research question and inclusion criteria have been modified in our manuscript. Suggestions from the recommended papers on the methodology of systematic reviews of economical topics have been read, integrated in the systematic review protocol and cited. We also added a relevant point to the initial strengths and limitations.

R2.4. Study design (page 6):

Guidelines and checklists are also relevant to add to the list.

(Full) economic evaluations are will provide information that is more relevant and are therefore important studies to consider for this review. Observational and interventional studies are less relevant to include.

Authors' Response	We agree, we added available guidelines and checklists as well as (full) economic evaluations to the study designs for inclusion to our "methods" section. Further we added where we will look for guidelines, checklists and value frameworks to the methods section.
---

	We will create a second data extraction sheet specifically for the characteristics of applied economic evaluations, which will be used for the data synthesis of the economic evaluations identified.
Actions	The study designs for inclusion have been modified accordingly. A second data extraction sheet for applied economic evaluations has been added to the appendix.

R2.5. Information sources and search strategy (pages 6-7) Besides searching for relevant studies in major databases, reports of HTA bodies are also be a relevant sources for finding information related to this topic (like NICE, <http://guidance.nice.org.uk/> and Canadian Agency for Drugs and Technologies in Health (CADTH), <https://www.cadth.ca>).

Searching for relevant citations (studies) can also be done by checking the references in known publications or by searching for additional studies that are cited in articles known to be relevant (such as Web of Science), this is also an essential step to find all relevant evidence.

Authors' Response	We thank the reviewer for the suggestions. We will screen relevant websites (e.g. standard health economic evaluation associations (e.g. ISPOR, iHEA), the resources of the members area of the International Collaborative for Best Care for the Dying Person, as well as reports of HTA bodies (e.g. NICE) which potentially provide economic guidelines, checklists and value frameworks for conducting economic evaluations in the palliative or end of life care settings. We agree with the reviewer in regards to the reference checking. As already stated in the manuscript, we indeed plan to also include relevant references, which are referred to in the identified articles.
Actions	The section on "information sources and search strategies" has been modified accordingly.

R2.6. Search syntax (page 7 and appendixes)

Please specify if peer reviewed, search strings were used. They to use them as much as possible; see <https://sites.google.com/a/york.ac.uk/issg-search-filters-resource/>.

Adapt search strings using the suggestions for PICO and eligibility criteria.

Where the search strategies already pilot tested?

Please specify how details of searches will be recorded.

Authors' Response	In collaboration with an information specialist we reviewed different published search filters for palliative care and health economics (e.g. CareSearch, Starr et al. (2019), Rietjens et al. (2019), Gomes et al. (2016), SIGN filter, Glanville (2018), CADTHS Database Search Filters (2019), McKinlay et al. (2006), Sassi (2002)). Simply applying one of these search filters was too insensitive for our case and revealed a too high number of irrelevant hits. Therefore, we explored different combinations of search terms and specifications and tested their sensitivity in regards to our research question by screening the first 100 articles for each adaption. In the end we developed a search strategy based on the examples of Rietjens et al.(2019) and Gomes et al. (2016), trying to include all relevant articles and still keeping the screening process feasible. We conducted another round of sensitivity testing of our search string, which revealed that the developed search string seems to be most appropriate for our scope, so we only had to conduct minor adaptations. The search string was now rechecked in light of the adapted PICO framework. It turned out that since we did not set any disease, context or outcome limitations in the first place and search terms for economic evaluations were included from the beginning, the
--

	initial search string remains the most appropriate and no further modifications are needed. The stepwise development process of the search will be saved and documented in doc format (OVID) and csv format (EMBASE). Further, the searches will be permanently saved on the accounts of the search databases. All modifications can be saved and retrieved for potential later updates. A flow diagram, which will be added to the manuscript, will transparently show the results of the searches as well as the inclusion/exclusion flow.
Actions	Details on the development process of the search string have been added to the “search syntax” subsection of the “methods” section.

R2.7. Data management (page 8)

Describe how data extraction sheets will be handled and which programs will be used for it (eg word, excel).

Authors’ Response	Excel sheets (one for studies on methodological aspects, one for applied economic evaluations) are planned to be used for the documentation of the extracted of information from the included articles. For illustration we included the data extraction data elements in appendix 3 of the systematic review protocol. The information from the data extraction sheets of studies on methodological aspects will be used for the narrative synthesis of the identified information on methodological problems and challenges per theme. For each theme (e.g. outcomes, costs) the identified information in regards to the described methodological challenge, the potential impact of the problem on results of economic evaluations and if identified, also the potential solution suggested in the literature will be subsequently summarized in a narrative way and recommendations will be formulated. The information collected in regards to study and patient characteristics of applied economic evaluations will also be narratively summarized. Further, an overview of methodological aspects, e.g. included outcomes and cost categories will be developed. See also our responses to R2.9.
Actions	Details of the data extraction sheets have been added to the “methods” section.

R2.8. Selection process (page 8)

How will selection of title and abstract will be done when no abstract is available? For instance for editorial, comments, or guidelines. Please specify.

Authors’ Response	In case of the absence of an articles title/abstract, the full text of the article will be directly screened for potential inclusion.
Actions	This information has been added to the section “selection process” in the manuscript.

R2.9. Data items and appendix 3

General aspects

The data extraction sheet (appendix 3) is very brief and contains no explanation/description of the items. This sheet could be improved by pilot testing it for a few studies. A picklist displaying varies answers in excel could decrease workload and could also improve consistency of answers.

Specific aspects

- *Not all studies will be published in a “journal” please add more answer options.*

- *Please adapt and add more details, in item “study design” (e.g. economic evaluations (EEs), guideline and checklist). Aspects of EE described are only applicable for EEs.*
- *Add open text space for all EE items. Information on: full or partial EE, Incremental cost-effectiveness analyses (ICERs) and analyses of uncertainty are also relevant to record.*
- *What are relevant items for guidelines, editorials, reviews and all other type of publications selected?*
- *Add more details/synonyms and examples for methodological problems/challenges, problems, concerns, recommendation and solutions.*
- *For the most items, it would be also relevant to add a space in the sheet, for recording the specific page or section where the information was found by the reviewer in the document.*
- *Finally, it is unclear why “Type of EE”, “resource use” and “ethical implications” are bold.*

Authors’ Response	We thank the reviewer for the careful revision of our data extraction sheet. The authors will use Excel files, containing drop down fields when appropriate (e.g. ‘choosing type of economic evaluation, study design) for free text fields for open questions (e.g. methodological problem described’, ‘potential solution suggested in study’) . The data extraction sheet in Appendix 3 has been created in Word for illustration only. The data extraction sheets have been pilot tested by the authors on five studies, which revealed that all items in the data extraction sheet are clear and data extraction based on them is possible. Nevertheless, we are aware that with such a broad review it may become necessary to modify the sheets during the data extraction process. We plan to do a formal round of revision of the data extraction sheets based on the reviewers’ experience after the first 10 studies have been processed. If a further division of the ‘themes’ section deems necessary/feasible, we will implement this step at that point. By clustering information on “outcomes” level, we will also be able to organize the identified data further individually and not have to decide a priori whether the information belongs to the one or the other outcome. The current themes, which we have been selected in regards to methodological aspects and challenges of conducting economic evaluations in the palliative and end of life care settings should be applicable irrelevant of the study design. Also in case of guidelines, editorials, reviews, etc. it will be possible to score to which theme its content refers to. In addition, we added the “other” section, as there might be other themes, we will come across which have not been listed in the data extraction sheet. However, we added a second data extraction sheet specifically for applied economic evaluations, which will allow to abstract characteristics in regards to the study design and patient population, as well as methodology applied. In order to keep the format data extraction sheet manageable, we would prefer to not extend the title of each section. The authors have developed each category title and know what they are referring to, which has also been confirmed in the pilot testing phase. Synonyms for “methodological problems” have now been added in the systematic review protocol for the reviewers. In the systematic review, information on the items in the data extraction sheet will be added. We agree with the reviewer that a data item to capture the page number on which information has been found should be added to the data extraction sheet. We also corrected the ‘type of EE”, “resource use” and “ethical implications” writing, which were bold by error.
Actions	The data extraction sheet has been modified as described above. The reference type has been added, the name of the journal has been extended to other types of

	publications, more options have been added to the “study design” section. Further, an additional field to capture the page number of the information identified has been added. The font of ‘type of EE’, “resource use” and “ethical implications” have been modified. A second data extraction sheet for applied economic evaluations has been added to Appendix 3 of the manuscript.
--	---

R2.10. Assessment of methodological quality (page 9-10) *Quality assessment of EEs or guidelines is very relevant for this review as it gives an indication of the methodological quality of the studies discussing the methodological challenges, which are the topic of interest for this review. Low quality studies/guidelines could have other challenges as good quality ones. On page 11, row 19 it is mentioned that the quality of evidence will be assessed, please explain how this will be done if you do not check the quality of the selected studies/guidelines (all types).*

Authors’ Response	We agree with the reviewers and we have now added formal quality assessment criteria to our data extraction sheet. As we will include diverse types of studies, we will use various quality appraisals for the different study designs, such as: such as NICE Quality appraisal checklists, CHEERS checklist or PRISMA checklist. See also our responses to R1.9 and R1.10.
Actions	The text in the manuscript has accordingly been modified. A section has been added to the data extraction sheet allowing to add the rating of the study’s quality appraisal.

R2.11. Funding statement (page 13) *Please state more clearly the name of the project.*

Authors’ Response	We agree.
Actions	The full name of the project has been added to the funding statement.

R2.12. Reference list

For improvement of this protocol, I would recommend to use the following paper series.

- 🔗 <https://doi.org/10.1080/14737167.2016.1246960>
- 🔗 <https://doi.org/10.1080/14737167.2016.1246962>
- 🔗 <https://doi.org/10.1080/14737167.2016.1246961>

Authors’ Response	We thank the reviewer for these very useful paper series. Based on this paper series, we expanded our protocol with details (for instance, the transferability assessment, external expert review), which we have not provided yet and referenced the 5-step approach, which we overall followed, to the extent it was applicable for our systematic review.
Actions	We expanded various sections in the protocol and added more details on the methodology of the different steps of our review, reviewed and cited the relevant parts of the recommended paper series.

Reviewer #3 -----

R3.1. Generally, this is a well written and well-structured manuscript describing a systematic review protocol. The title adequately describes the study and the abstract is a fair summary of the study. The introduction clearly states the objectives of the study.

Overall, the methods chosen appear relatively robust and appropriate for the stated objectives based on the information provided. However, given the review’s aims, I think “economic evaluations testing the efficiency of an intervention or economic evaluations in the palliative care settings” should be included in the review, as the discussion sections of such articles will provide useful information on methodological aspects and challenges of conducting economic evaluations in the palliative and end of life care setting. Omitting such studies would be a major limitation of the review.

Authors’ Response	We thank the reviewer for this comment. We agree that the discussion sections of economic evaluations could potentially provide useful information on methodological aspects and therefore, we agree to include them.
Actions	Economic evaluations have been added to the list of study designs, which will be considered for inclusion in our “methods” section. We will screen the applied economic evaluations for methodological aspects and challenges described, and in addition also collect information on the study characteristics and patient population (e.g. cost categories considered and outcomes assessed). For this purpose a second data extraction sheet has been created (Appendix 3, data extraction sheet for applied economic evaluations). See also our responses to R2.4.

R3.2. On a more minor note, there are a number of published and publicly available search filters for identifying palliative care research (see CareSearch as example <https://www.caresearch.com.au/caresearch/tabid/377/Default.aspx>). Why not use one of these validated search filters?

Authors’ Response	We agree with the reviewer that it’s recommended to use available search filters, especially validated ones, as much as possible. Therefore we reviewed existing search filters (such as CareSearch, Starr et al. (2019), Rietjens et al. (2019), Gomes et al. (2016), SIGN filter and Glanville (2018)), together with an information specialist. In order to increase the sensitivity for our specific research aim we tried different combinations of search terms and specifications followed by a pilot screening of the first 100 articles after each adaption. We developed a search strategy based on the examples of Rietjens et al.(2019) and Gomes et al. (2016) and conducted an additional round of sensitivity testing of our search string, which revealed that the developed search string seems to be most appropriate for our scope. See also our responses to R2.6.
Actions	Details on the development process of the search string has been added to the “search syntax” subsection of the “methods” section.

R3.3. Please clarify whether the two reviewers are independently screening the titles and abstracts (first screening round). How will the kappa statistic inform the process, e.g. what if the kappa value is low?

Authors’ Response	We agree with the reviewer that this was not completely clear from our protocol. Yes, two reviewers will independently screen the titles and abstracts. A pilot round of the first 100 identified articles will be conducted for which the kappa value will be calculated. In case this reveals to be low, the inclusion and exclusion criteria will be discussed again
---

	among the two researchers, as well as the articles, which they did not agree on, in order to rule out any potential misinterpretation of the screening criteria. By this we hope to increase the level of agreement between the reviewers and a high kappa value in the full screening process. In case the kappa value turns out to be low after the full screening process, the reviewers will have to do an in depth review round of their disagreements and consult a third reviewer.
Actions	We clarified this information in the “methods” section of our manuscript.

R3.4. Whilst the results of this systematic review should provide interesting findings, important information on methodological aspects and challenges of conducting economic evaluations in the palliative and end-of-life care setting may be missed due to the exclusion criteria.

Authors' Response	We agree with the concern of the reviewer and have therefore modified our inclusion criteria and will also include and screen economic evaluations themselves for their characteristics, and potential methodological aspects and challenges described.
Actions	Economic evaluations have been added to the list of study designs, which will be considered for inclusion in our “methods” section.

VERSION 2 – REVIEW

REVIEWER	Gardiner, Clare University of Sheffield
REVIEW RETURNED	18-Feb-2020

GENERAL COMMENTS	Thank you for addressing the comments so comprehensively, the paper is much improved and I look forward to seeing the results of this important systematic review when it is published. There are a few minor issues to address but these aside I would be happy to recommend it for publication. P3 – typo quiet for quite P3 – define iLIVE PP advisory board? Is this patients and public? P4 – The following statement is inaccurate “the ultimate aim of interventions in the palliative or end of life care settings is not to improve quality of life”. According to WHO and other definitions palliative care is very much concerned with improving quality of life. You are correct in saying that palliative care is not concerned with the extension of life and I would focus on this. P5 – you claim that the review will help generate guidance irrespective of geographic region. Please be mindful that the guidance you produce may not be relevant for LMIC's given the vast differences in palliative care provision and funding in these regions. P5 objective 1 - should this be 'gender OR geographic region rather than 'gender nor their geographic region' The English language is generally good but may also need some review for minor grammar and style inaccuracies.
---

REVIEWER	Ghislaine van Mastrigt Maastricht University, The Netherlands
-----------------	--

REVIEW RETURNED	02-Mar-2020
-------------

GENERAL COMMENTS	I think you did a really good job in improving this study protocol. I am satisfied with the changes made to the manuscript. I have only one comment to make: -cross check the studies designs in table 1 and studies listed in "design" section of the method section (page 7), please adapt where necessary
---

VERSION 2 – AUTHOR RESPONSE

Reviewer #1 -----

R1.1 Thank you for addressing the comments so comprehensively, the paper is much improved and I look forward to seeing the results of this important systematic review when it is published. There are a few minor issues to address but these aside I would be happy to recommend it for publication.

Authors'

Response We would like to again thank Reviewer 1 for her comments, which helped to improve this protocol.

Actions /

R1.2

P3 – typo quiet for quite

Authors'

Response Thank you for noticing.

Actions The typo has been corrected.

R1.3

P3 – define iLIVE PP advisory board? Is this patients and public?

Authors'

Response Yes, the iLIVE PP advisory board is currently set-up and will consist of patients and public advisors.

Actions Details on the iLIVE PP advisory board have been added in the manuscript.

R1.4

P4 – The following statement is inaccurate “the ultimate aim of interventions in the palliative or end of life care settings is not to improve quality of life”. According to WHO and other definitions palliative

care is very much concerned with improving quality of life. You are correct in saying that palliative care is not concerned with the extension of life and I would focus on this.

Authors'

Response We agree with the Reviewer that this statement could be misleading and the aim of palliative or end of life care should be specified in our manuscript.

Actions This statement has been refined.

R1.5

P5 – you claim that the review will help generate guidance irrespective of geographic region. Please be mindful that the guidance you produce may not be relevant for LMIC's given the vast differences in palliative care provision and funding in these regions.

Authors'

Response We agree with the Reviewer on the caution which is required in regards to the transferability between geographic regions. In our search strategy, we did not restrict ourselves to any specific geographic region. Yet, in the data analysis phase, we will, of course, consider from which geographic region the identified evidence and guidance is coming from and in how far it can be generalized for other regions. We will conduct a formal transferability assessment as far as possible. Nevertheless, as already stated in the manuscript, a separate review tailored to low and middle-income settings may become necessary if the current review identifies relevant cultural/spiritual/preference aspects important also for economic evaluations.

Actions /

R1.6

P5 objective 1 - should this be 'gender OR geographic region rather than 'gender nor their geographic region'

The English language is generally good but may also need some review for minor grammar and style inaccuracies.

Authors'

Response The Reviewer is right, this is a mistake.

Actions The mistake has been corrected and the manuscript has anew been reviewed for spelling and grammar errors.

Reviewer #2 -----

(R2.1) Dear authors,

I think you did a really good job in improving this study protocol. I am satisfied with the changes made to the manuscript.

I have only one comment to make:

-cross check the studies designs in table 1 and studies listed in "design" section of the method section (page 7), please adapt where necessary

Authors'

Response We want to thank reviewer 2 again for her comments to improve this protocol. We now double-checked the study designs listed in the Methods section for inconsistencies.

Actions Table 1 has been modified to match the information in the "design" section of the Methods section.